# Experimental Study and Mathematical Modeling of a Glyphosate Impedimetric Microsensor Based on Molecularly Imprinted Chitosan Film

**Fares Zouaoui [1,2], Saliha Bourouina-Bacha [2], Mustapha Bourouina [2], Albert Alcacer [3], Joan Bausells [3], Nicole Jaffrezic-Renault [1,*], Nadia Zine [1] and Abdelhamid Errachid [1,*]**

[1] University of Lyon, Institute of Analytical Sciences, 69100 Villeurbanne, France; fareszou06@gmail.com (F.Z.); nadia.zine@univ-lyon1.fr (N.Z.)
[2] Faculty of Technology, University of Bejaia, Bejaia 06000, Algeria; lgebej@yahoo.fr (S.B.-B.); bouryas@yahoo.fr (M.B.)
[3] CSIC, CNM, IMB, Campus UAB, 08193 Barcelona, Spain; albert.alcacer@imb-cnm.csic.es (A.A.); joan.bausells@imb-cnm.csic.es (J.B.)
* Correspondence: nicole.jaffrezic@univ-lyon1.fr (N.J.-R.); abdelhamid.errachid-el-salhi@univ-lyon1.fr (A.E.)

**Abstract:** A novel impedimetric microsensor based on a double-layered imprinted polymer film has been constructed for the sensitive detection of the herbicide, glyphosate (GLY), in water. It is based on electropolymerized polypyrrole films, doped with cobaltabis(dicarbollide) ions ($[3,3'\text{-Co}(1,2\text{-}C_2B_9H_{11})_2]$), as a solid contact layer between the gold microelectrode surface and the molecularly imprinted chitosan film (CS-MIPs/PPy/Au). Electrochemical Impedance Spectroscopy (EIS) was used for the characterization of the CS-molecular imprinted polymers (MIPs)/PPy/Au in the presence of GLY concentrations between 0.31 pg/mL and 50 ng/mL. Experimental responses of CS-MIPs/PPy/Au are modeled for the first time using an exact mathematical model based on physical theories. From the developed model, it was possible to define the optimal range of the parameters that will impact the quality of impedance spectra and then the analytical performance of the obtained microsensor. The obtained microsensor shows a low detection limit of 1 fg/mL (S/N = 3), a good selectivity, a good reproducibility, and it is regenerable.

**Keywords:** Glyphosate; polypyrrole; molecularly imprinted polymer; chitosan; electrodeposition; impedimetric micro-sensor; mathematical modeling

---

## 1. Introduction

Glyphosate (GLY) is an effective systemic weed herbicide that was introduced for weed control in agricultural production fields around the world. Glyphosate is very resistant to degradation due to the inert C-P bond in the molecule [1]. GLY was frequently detected in rain and air and it is a major pollutant of rivers and surface waters. GLY can contaminate organisms, including humans, food, and ecosystems [2,3], which suggests its potential risks. Increasing studies have shown that glyphosate-based herbicides show neurotoxicity, cytotoxicity, and endocrine toxicity [4]. Therefore, development of methods for GLY detection is attracting more interest. Numerous analytical methods have been reported in the current literature, such as gas chromatography, high performance liquid chromatography, capillary electrophoresis [5], Mass spectrometry [6], resonance spectrometry, fluorescent spectrometry, an enzyme-linked immunoassay, and electrochemical sensors [7].

Molecular imprinted polymers (MIPs) are techniques based on an artificial recognition of target molecules. MIPs are prepared with a reaction mixture composed of a template (target molecule) and a functional monomer. During the polymerization, a complex is formed between the matrix and

the functional monomer, and then the complex is surrounded by a crosslinking agent. MIPs can be prepared according to a number of approaches. The template can be bound to the functional monomer by reversible covalent bonds or non-covalent bonds [8].

MIPs are techniques with a low production cost. Moreover, they have important properties such as physical strength, robustness, resistance to high pressures and temperatures, and increased inertness to various chemicals [9,10]. In addition, the main advantages of MIPs are their high affinity and their selectivity for the target molecule [11]. The choice of a functional monomer is of major importance considering their ability to provide complementary interactions with the template molecules.

Chitosan is obtained from chitin, extracted from shrimp shells, after the acetylation process, which allows the partial elimination of acetyl groups. Chitosan has three types of reactive functional groups, an amino group, and primary and secondary hydroxyl groups at the C-2, C-3 and C-6 positions, respectively (Figure 1). The presence of reactive functional groups allows numerous chemical modifications through numerous chemical reactions such as alkylation, reaction with aldehydes and acetones, grafting, H-bonding with polar atoms, and crosslinking [12]. Many successful electrochemical sensors based on chitosan-MIPs was developed for different templates [13].

**Figure 1.** Chemical structure of chitosan.

Polypyrrole (PPy) is one of the most promising materials for many applications because of its good chemical and thermal stability, facile synthesis, high conductivity, and its environmentally friendly properties [14,15]. PPy has attracted much attention in many electrochemical applications such as sensors and biosensors. It was used as a solid internal contact between the metal and the ion-selective membrane to facilitate the charge transfer at the substrate/film interface [16]. PPy admits a porous structure with a large specific surface area. This property is an asset for several applications because of the high charge/discharge rate [17]. Many doping anions can be incorporated into PPy films. The cobaltbis(dicarbollide) anion $[3,3'-Co-(1,2-C_2B_9H_{11})_2]^-$ was established as an ideal hydrophobic anion for ion extraction through an ion-pair mechanism [18]. The resulting PPy polymer doped with cobalt bis(dicarbollide) anion showed enhanced thermal stability and a dramatic enhancement of its overoxidation threshold, which demonstrated a great improvement of the electrical characteristics of film [19].

The response of 'miniaturized developed' sensors is closely related to the presence of specific molecular imprints on the biopolymer film's surface of the working micro-electrode, and also related to factors inherent to the structure of the sensor itself and its mode of operation. Explaining the operation in the depths of this sensor requires an in-depth knowledge of all the parameters involved in such a design. Modeling is a simplified representation of a real physical system or phenomenon, making it possible to reproduce its functioning, to analyze it, to explain it, and to predict certain aspects of it. Modeling is a tool, which helps understand the intrinsic mechanisms of these analytical instruments. It makes it possible to find relationships between the variables and the parameters that are considered to influence the metrological characteristics of these sensors such as the resistance of the solution, the resistance to charge transfer, or geometry. It is essential for optimization before a device or process goes to market. A reliable model is one that simulates a sensor in real conditions. In addition, there arises the problem of determining the parameters, which intervene in the equations of the model.

In this study, a novel MIP electrochemical impedance spectroscopy (EIS) sensor has been constructed for the sensitive detection of GLY. The gold microelectrode surface was coated with a functional conducting polymer doped with cobaltbis(dicarbollide) anion ([3,3'-Co(1,2-$C_2B_9H_{11}$)$_2$]) via electrochemical polymerization by cyclic voltammetry (CV). Then, the chitosan sensitive membrane was electrodeposited on the conductive polymer layer. Electrochemical Impedance Spectroscopy (EIS) was used for the characterization of the CS-MIPs/PPy/Au in the presence of GLY concentrations. A mathematical model based on physical theories is developed to analyze the data obtained experimentally. The analysis of the observed impedance metric response leads to the estimation of the microscopic parameters of the sensors. The validation of the model is obtained by comparing the experimental data to the theoretical impedance model.

## 2. Materials and Methods

### 2.1. Reagents

Chitosan (CS, degree of deacetylation 80.0%–95%, molecular weight $M_w$ = 250 KDa), acetic acid (purity 99.7%), methanol (purity 99.9%), sulfuric acid (purity 95%), sodium hydroxide (NaOH), pyrrole (purity 99%), acetonitrile (purity 99.9%), and glyphosate (GLY) were obtained from Sigma Aldrich. Cesium Cosane (Cs [3,3'-Co(1,2-$C_2B_9H_{11}$)$_2$]) was purchased from Katchem spol. s.r.o. It was used as doping agent for the preparation of a conductive-polymer intermediate solid layer between the sensitive membrane and gold substrate. Gluphosinate-ammonium (GLU), chlorpyrifos (CHL), and phosmet (PHO), used as interfering pesticides, were provided by Sigma Aldrich. All these chemicals were of an analytical reagent grade. Experiments were carried out at ambient temperature.

### 2.2. Apparatus

All electrochemical techniques were carried out using a multi-channel potentiostat (Biologic-EC-Lab VMP3) analyser. All measurements were carried out using a transducer fabricated at the National Center for Microelectronics (CNM), CSIC, Spain. It holds an array of four bare-gold working microelectrodes (WE) (surface area: 0.64 mm$^2$), one counter microelectrode (CE) (surface area: 0.13 mm$^2$), and two Ag/AgCl reference microelectrodes (RE) (surface area: 1.37 mm$^2$), connected at the same time and controlled by a personal computer (see transducer in Figure 2A). The pH of solutions was measured using a pH-meter: Mettler Toledo FE20/EL20. Scanning electron microscopy (SEM) micrographs were obtained using a FEI Quanta FEG 250 (University of Lyon 1, France).

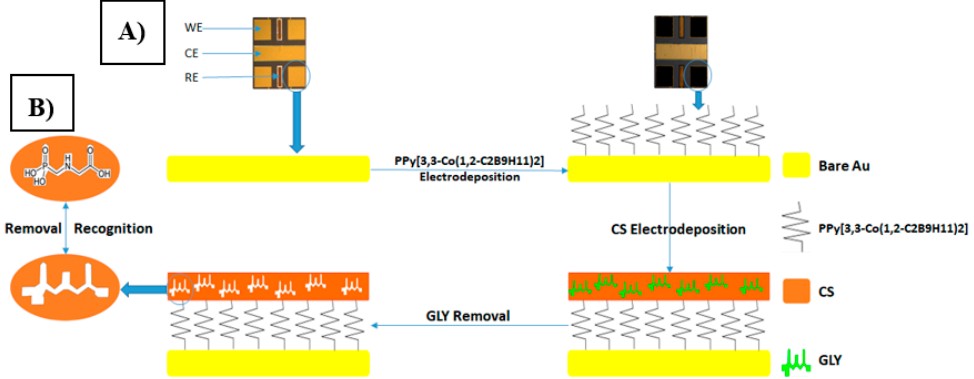

**Figure 2.** (**A**) Chip holding an array of four bare-gold working microelectrodes (WE) s = 0.64 mm$^2$, one counter microelectrode (CE) s = 0.13 mm$^2$ and two Ag/AgCl reference microelectrodes (RE) s = 1.37 mm$^2$. (**B**) Preparation of molecularly imprinted chitosan film (CS-MIPs/PPy/Au) and its recognition for glyphosate (GLY).

### 2.3. Preparation CS-MIPs/PPy/Au Sensor

**Electrode preparation:** The Au microelectrode was rinsed with ethanol and deionized water in ultrasound for 10 min and then exposed to the UV/ozone for 30 min.

**PPy[3,3'-Co(1,2-C$_2$B$_9$H$_{11}$)$_2$] electrodeposition step:** A solid contact layer of polypyrrole conductive polymer doped with [3,3'-Co(1,2-C$_2$B$_9$H$_{11}$)$_2$]$^-$ anion was galvanostatically grown by electrochemical polymerization onto gold microelectrodes. The solution was made of 0.035 M of Cs[3,3'-Co(1,2-C$_2$B$_9$H$_{11}$)$_2$] and 0.1 M of pyrrole in acetonitrile 1 wt.% in water. The electrochemical polymerization was carried out by applying five potential sweep cycles between −0.6 V and 1.2 V, at a scan rate of 100 mV/s by means of cyclic voltammetry (CV) (Figure 3A). After polymerization, the microelectrodes were rinsed with deionised water and dried under nitrogen flow [20].

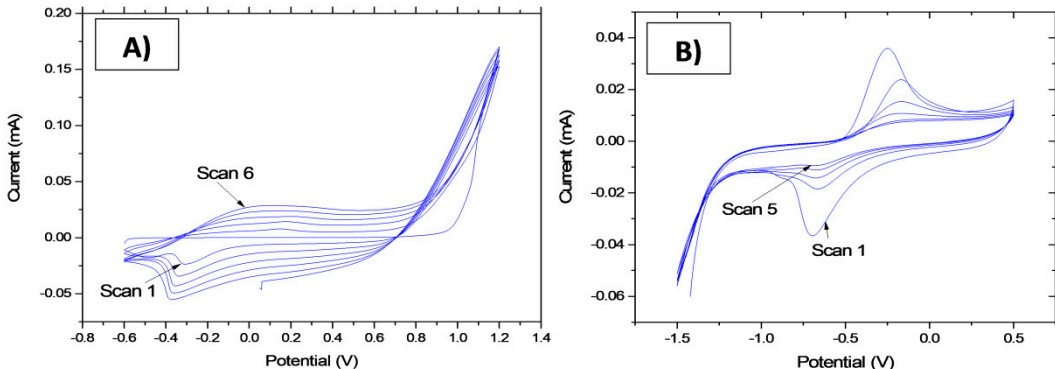

**Figure 3.** (**A**) Cyclic voltammogram of PPy[3,3_-Co(1,2-C$_2$B$_9$H$_{11}$)$_2$] electrodeposition on a microelectrode surface. (**B**) Cyclic voltammogram of CS-MIPs electrodeposition on PPy film.

**CS-MIPs electrodeposition step:** The experimental conditions were optimized in our previous work [21]. In total, 1 g of CS powder was dissolved in 100-mL 0.1 M acetic acid and ultra-sonicated for 6 h at room temperature. The GLY-CS suspension system was prepared by dispersing 10 mg GLY into a 10-mL chitosan solution with a template/monomer ratio equal to 1/10, and mixed for 2 h to promote interactions between glyphosate and chitosan. Then, the pH value of solution was adjusted to be 5, using 0.1 M NaOH. GLY-CS suspension was deposited using cyclic voltammetry (CV) for five scans in the range of −1.5 to 0.5 V at a scan rate of 80 mV/s (Figure 3B). After electrodeposition, the microelectrodes were rinsed with deionized water and dried using nitrogen.

**Cross-linking and template removal steps.** They consisted of incubating the (GLY + CS)/Au sensor in 0.5 M H$_2$SO$_4$ solution for 1 h, which was then followed by an incubation in acetic acid/methanol solution (1:1, v/v) for 30 min to remove the GLY template.

Thus, an electrochemical sensor based on electropolymerized polypyrrole films, doped with cobaltabis (dicarbollide) ions ([3,3'-Co(1,2-C$_2$B$_9$H$_{11}$)$_2$]), as a solid contact layer between the gold microelectrode surface and the molecularly imprinted chitosan membrane was developed. The process is shown in Figure 2B. The non-imprinting polymer sensor (CS-NIPs/PPy/Au) followed similar steps as the CS-MIPs/PPy/Au electrodes with only one major difference. The CS mixture for the NIPs did not contain GLY. Lastly, sensors were stored at room temperature for further use.

### 2.4. Electrochemical Measurements

CS-MIPs/*PPy*/Au was immersed in water containing GLY with different concentrations (0.31 pg/mL to 50 ng/mL) for 30 min. Then, EIS (Initial potential E = 0.2 V. Higher Freq = 100 kHz, Lower Freq = 1 Hz) was used to characterize microelectrode surfaces and to investigate the charge transfer resistance of the film. EIS measurements were performed in ferro-ferricyanide with phosphate buffer saline solution (PBS). Cyclic voltammetry from 0 to 0.45 V at a scan rate of 80 mV/s was also used to characterize microelectrode surfaces.

## 3. Results and Discussion

### 3.1. Microsensors Characterization

#### 3.1.1. Electrochemical Characterization of CS-MIPs/PPy/Au

Polypyrrole doped with [3,3'-Co(1,2-C$_2$B$_9$H$_{11}$)$_2$] anion was grown onto gold substrate in order to improve charge transfer and adhesion properties at the interface. The use of conducting polymers as solid contact materials was proposed as a conductive interface between the chitosan-MIPs film and the metal substrate. Figure 4A,B show the electrochemical impedance spectroscopy diagrams and the cyclic voltammograms of gold microelectrodes after electropolymerization of PPy[3,3'-Co(1,2-C$_2$B$_9$H$_{11}$)$_2$] on the microelectrode, after electrodeposition of CS-MIPs onto a solid contact layer crosslinked with sulfuric acid, and after template removal of CS-NIPs/PPy/Au. It can be seen from the EIS (Figure 4A,B) that the impedance presented a clear decrease after electropolymerization of the PPy (R$_{ct}$ = 68.7 Ω), compared to the bare electrode (R$_{ct}$ = 342.4 Ω) (Figure 4A, a) due to the presence of the conductive polymer that enhances the electric-charge-transfer properties of the electrode. Then, the electrodeposition of CS on the Au/PPy surface resulted in a clear increase of the charge transfer resistance due to blockage of the surface by the chitosan film (R$_{ct}$ = 17,691 Ω) (Figure 4A, c). If we compare with the NIP film (R$_{ct}$ = 3555 Ω) (Figure 4A, e), a larger Re (Z) is observed for the MIP film, which would show that MIPs film is thicker than NIPs film. After extraction of the GLY template, a significant decrease in impedance has been observed (R$_{ct}$ = 991.2 Ω) (Figure 4A, d). Moreover, a lower charge transfer resistance is observed comparing to that of NIPs. This difference is due to the imprinted cavities, which promote the electron transfer.

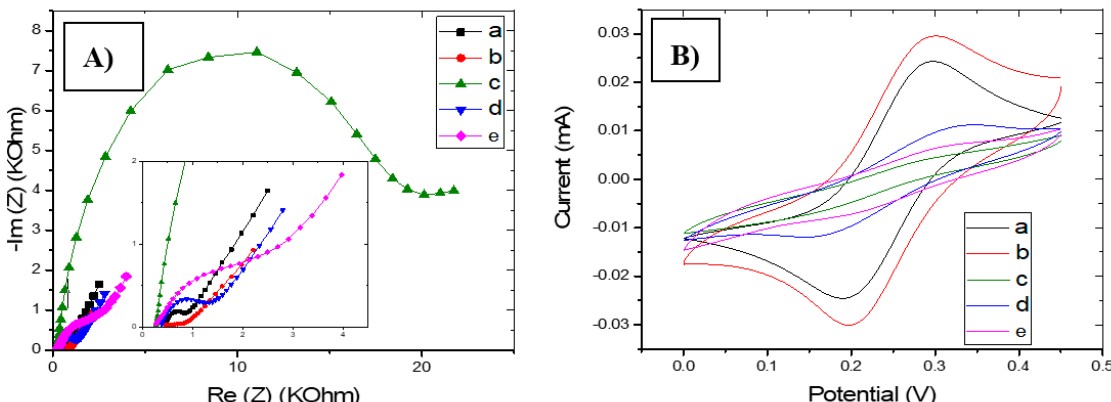

**Figure 4.** (**A**) Electrochemical impedance spectroscopy (EIS) of bare Au (a), PPy/Au (b), CS-MIPs/PPy/Au before extraction of the template (c), CS-MIPs/PPy/Au after extraction of the template (d), CS-NIPs/PPy/Au (e). in 5 mM [Fe(CN)$_6$]$^{3-/4-}$ and phosphate buffer saline solution (PBS), initial potential E= 0.2 V. Higher Freq = 100 kHz, Lower Freq = 1 Hz. (**B**) Cyclic voltammetry (CV) of bare Au (a), PPy/Au (b), CS-MIPs/PPy/Au before extraction of template (c), CS-MIPs/PPy/Au after extraction of the template (d), CS-NIPs/PPy/Au (e). in 5 mM [Fe(CN)$_6$]$^{3-/4-}$ and PBS from 0 to 0.45 V at a scan rate of 80 mV/s.

This is further confirmed by cyclic voltammetry (CV), which was used to assess the electron transfer rate for the modified working electrode (WE). This was done by submerging the modified WE in ferro-ferricyanide and running the CV from 0 to 0.45 V at a scan rate of 80 mV/s. It can be seen from the cyclic voltammogram (Figure 4B) how redox peaks increased in terms of intensity current after the electrodeposition of PPy[3,3'-Co(1,2-C$_2$B$_9$H$_{11}$)$_2$] layer (the anodic current I$_a$ increases from 24 µA to 30 µA) (Figure 4B, b). Then it was decreased after electro-polymerization of the chitosan-MIP film (I$_a$ = 5.6 µA) (Figure 4B, c) diminished the electric charge transfer properties of the microelectrode. Comparing to the NIP film (I$_a$ = 10 µA), (Figure 4B, e), the lower value of redox peaks is observed

for the MIP film, which shows that the latter is thicker than the NIP film. After extraction of the GLY template, a significant increase in redox peaks has been observed ($I_a$ = 7.5 µA) (Figure 4B, d) due to the opening of the imprinted cavities.

### 3.1.2. Surface Morphology

Scanning electron microscopy (SEM) was employed to investigate the surface morphologies of bare electrode, PPy/Au, CS-MIPs/PPy/Au, and CS-NIPs/PPy/Au (Figure S1). The morphology of the microelectrode modified with PPy (Figure S1B) is completely different when compared with the surface of the bare gold (Figure S1A), which indicates the success of the electro-polymerization of the conductive layer. After electro-polymerization of the chitosan onto a solid contact layer, another layer has appeared on the surface of the microelectrode, which further changes the morphology and confirms the deposition of the CS (Figure S1C). Morphologies of the MIP (Figure S1C) and of the NIP films (Figure S1D) present a globular aspect. It is difficult to differentiate between them.

### 3.2. Electrochemical Responses of the CS-MIPs/PPy/Au

EIS was employed for the quantitative detection of GLY. As shown in Figure 5, a gradual increase of EIS was observed with the increase in GLY concentration, indicating a correlation between GLY concentration and the impedance of the CS-MIP/PPY/Au microelectrode.

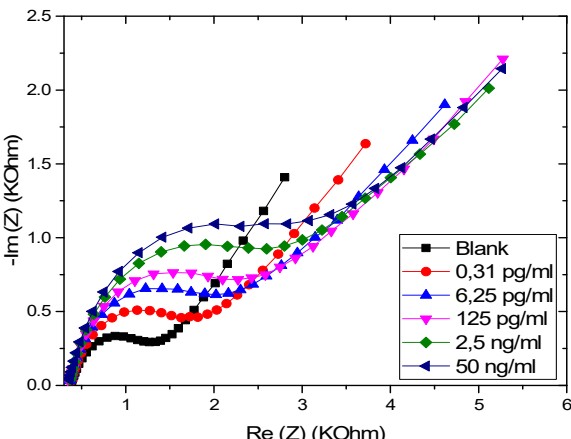

**Figure 5.** EIS of CS-MIPs/PPy/Au in 5 mM [Fe(CN)$_6$]$^{3-/4-}$ and PBS. Before pre-concentration and after incubation in different concentrations of glyphosate (GLY).

### 3.3. Modeling of the CS-MIPs/PPy/Au Microsensor

#### 3.3.1. Mathematical Model

The general diagram of the impedance spectrum obtained by spectroscopic measurement for the experimental cell is illustrated in Figure 6. The overall impedance spectrum is made up of two well-separated regions: From 100 kHz to 150 Hz corresponding to a semi-circle, which associates the charge transfer resistance in parallel with the double layer capacitance usually described by the Constant Phase Element (CPE). At high frequency (100 KHz), the intersection of the impedance curve with the abscissa axis makes it possible to determine the resistance $R_s$, which models the electrical conductivity of the solution due to mobile ions. From 150 Hz to 1 Hz, the linear part of the diagram corresponds to the Warburg impedance (diffusion of electroactive species) [22,23].

The response of the designed micro-sensors is marked by the variation of the semi-circles toward the concentrations of GLY incubated. This variation, according to the impedance spectra, is more significant in comparison with the variation in the Warburg diffusion.

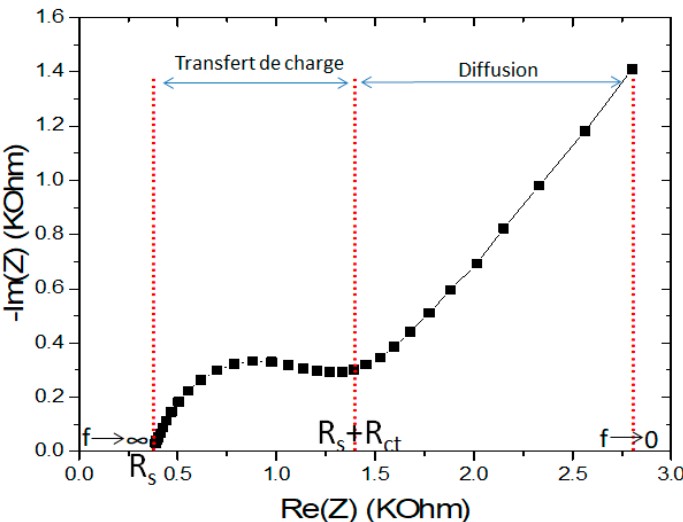

**Figure 6.** Nyquist plot for electrochemical impedance response of CS-MIPs/PPy/Au.

To simplify the equations of the physical model, the response of the sensor is modeled with a series combination of the resistance of the solution $R_s$ and the electron resistance transfer $R_{ct}$ placed in parallel with CPE. The equivalent circuit is shown in Figure 7.

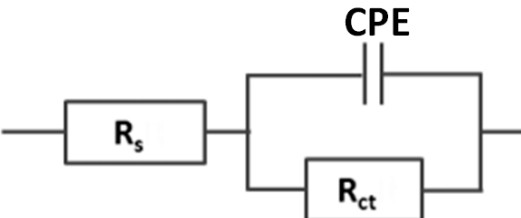

**Figure 7.** An equivalent electric circuit composed of two resistors and constant phase elements ($R_s$+CPE/$R_{ct}$).

As defined in the literature, the CPE represents many elements such as the inhomogeneity of the surface, the inhomogeneity of the charge distribution, and of the coupling between the faradic and capacitive currents. It can, therefore, be expected that a better fit for real systems will be obtained by using the CPE as a replacement for the Cdl capacitance [24]. To model this behavior, a fractional element CPE is proposed and expressed as follows [25].

$$Z_{CPE} = \frac{1}{Q(j\omega)^n} \tag{1}$$

$$j = \sqrt{-1} = \cos\left(\frac{\pi}{2}\right) + j\sin\left(\frac{\pi}{2}\right) \tag{2}$$

With $Q$ (F $s^{(n-1)}$) representing the CPE coefficient, $-1 \leq n \leq 1$ is the correction factor, $\omega = 2\pi f$, where f represents the frequency (Hz). Equation (1) can also be written as shown below.

$$Z_{CPE} = \frac{1}{Q\omega^n}\left[\cos\left(n\frac{\pi}{2}\right) - j\sin\left(n\frac{\pi}{2}\right)\right] \tag{3}$$

The global impedance equivalent to the circuit in Figure 7 is given by:

$$Z = R_s + \frac{R_{ct}\left(1 + R_{ct}Q\omega^n \cos\frac{n\pi}{2}\right)}{1 + (R_{ct}Q\omega^n)^2 + 2R_{ct}Q\omega^n \cos\frac{n\pi}{2}} - j\frac{R_{ct}^2 Q\omega^n \sin\frac{n\pi}{2}}{1 + (R_{ct}Q\omega^n)^2 + 2R_{ct}Q\omega^n \cos\frac{n\pi}{2}} \tag{4}$$

That can be broken down into:

$$\text{Re}(Z) = R_s + \frac{R_{ct}\left(1 + R_{ct}Q\omega^n \cos\frac{n\pi}{2}\right)}{1 + (R_{ct}Q\omega^n)^2 + 2R_{ct}Q\omega^n \cos\frac{n\pi}{2}} \tag{5}$$

$$-\text{Im}(Z) = \frac{R_{ct}{}^2 Q\omega^n \sin\frac{n\pi}{2}}{1 + (R_{ct}Q\omega^n)^2 + 2R_{ct}Q\omega^n \cos\frac{n\pi}{2}} \tag{6}$$

where Re (Z) is the real part of the impedance and −Im (Z) is the imaginary part of the impedance.

The reactions at the electrodes involve the redox couple Ferri/ferrocyanide in which ferricyanide is the oxidant and the ferrocyanide is the reducing agent. The electrochemical equation corresponding to this couple is as follows.

$$Fe^{III}(CN)_6^{3-} + e^- \leftrightarrow Fe^{II}(CN)_6^{4-} \tag{7}$$

In the case where the reactions at the electrodes are governed by the kinetics of electron transfer, the resistance to charge transfer $(R_{ct})$ is known as the opposition to the movement of the electrons. For $C_{Ox} = C_{Red} = C$, and, for a simple one-electron process ($n = 1$), $R_{ct}$ is given by Equation (8).

$$R_{tc} = \frac{RT}{F^2 A k^0 C} \tag{8}$$

where R: Ideal gas constant $(J \cdot mol^{-1} \cdot K^{-1})$, T: Temperature $(K^\circ)$, F: Faraday constant $(C \cdot mol^{-1})$, A: surface of the working electrode $(cm^2)$, and $k^0$: standard rate constant (cm/s).

The global impedance Z tends toward the resistance of the solution Rs when the frequency tends towards zero. $R_s$ is defined by Equation (9) [26,27].

$$R_s(\Omega) = \rho\frac{l}{A} \tag{9}$$

where $\rho$ is the resistivity of the solution ($\Omega$ cm), A is the surface area of the electrode $(cm^2)$, and $l$ is the coating thickness (cm).

The resistivity of the Ferri/ferrocyanide solution is calculated by the following relationships.

$$\rho = \frac{1}{\sigma} \tag{10}$$

$$\sigma = \sum_i q_i \lambda_i C_i \tag{11}$$

where $\sigma$ is the conductivity of the solution (s/m) that can be calculated from redox ion conductances [28,29], $q_i$ is the number of charges of the ion, $\lambda_i$ is the equivalent molar ionic conductivity ($\lambda(Fe^{III}(CN)_6{}^{-3}) = 10.09$ ms·m²/mol, $\lambda(Fe^{II}(CN)_6{}^{-4}) = 11.04$ ms·m²/mol), and $C_i$ is the concentration of the ion (mol/m³). The resistivity value determined is 2.687 $\Omega$ m.

### 3.3.2. Numerical Simulation

To be able to determine the parameters of this model from experimental data, the Matlab software was used to simulate the physical model. The temperature (T), the ideal gas constant (R), the resistivity of the solution ($\rho$), the Faraday constant (F), the surface of the electrode (A), the concentration (C), the angular velocity vector ($\omega$), the permissible error ($e_0$), the experimental Re (Z), and the −Im (Z) are used as input parameters.

Theoretical Re (Z) and −Im (Z) have been calculated in several iterations for each step of the membrane thickness (l), the speed constant (k°), the CPE coefficient (Q), and the correction factor (n). The variation interval of each parameter and the values of the input parameters are shown in Table 1.

**Table 1.** Parameters and variables used in the numerical simulation.

| Input Parameters | Value | Unit | Variables | Variation Range | Unit |
|---|---|---|---|---|---|
| T | 298 | K | $l$ | $[10^{-3}, 10^{-5}]$ | cm |
| F | 96485 | c·mol$^{-1}$ | k° | $[10^{-3}, 10^{-5}]$ | cm·s$^{-1}$ |
| R | 8.3145 | J·mol$^{-1}$·K$^{-1}$ | Q | $[10^{-5}, 10^{-8}]$ | s$^n$·Ω$^{-1}$ |
| A | 0.0064 | cm$^2$ | n | $[-1,1]$ | / |
| ρ | 268.7 | Ω·cm | | | |
| C | $5 \times 10^{-6}$ | mol·cm$^{-3}$ | | | |
| Ω = 2πf | f = 100 KHz→150 Hz | rad·s$^{-1}$ | | | |
| Re(Z) | / | Ω | | | |
| −Im(Z) | / | Ω | | | |

The values of various parameters are determined by minimizing the error between the experimental data and the simulated responses. This error is calculated according to the following relationship.

$$Error(e_1) = \sum \left| \frac{Re(Z)_{theoretical} - Re(Z)_{experimental}}{Re(Z)_{theoretical}} \right| \tag{12}$$

$$Error(e_2) = \sum \left| \frac{(-Im(Z))_{theoritical} - (-Im(Z))_{experimental}}{(-Im(Z))_{theoretical}} \right| \tag{13}$$

The iterations are stopped when $e_1, e_2 \leq e_0$ in this study and the tolerated error is $e_0 = 10^{-3}$. The general algorithm of the numerical program is given in Figure 8.

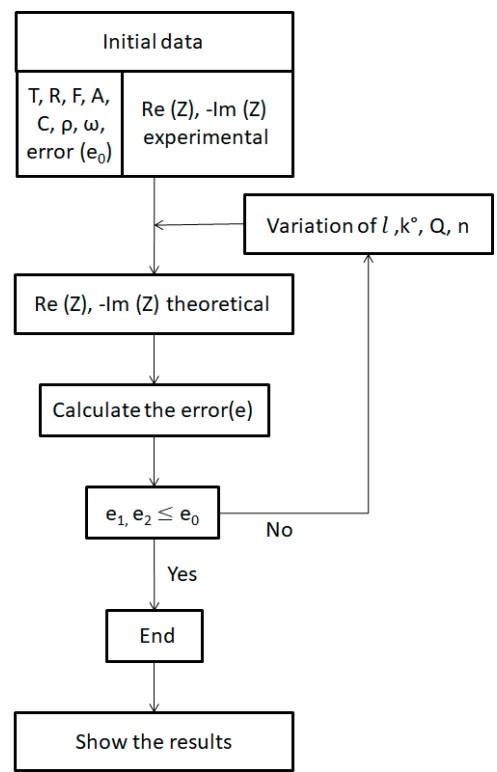

**Figure 8.** Diagram of the general programming algorithm.

### 3.3.3. Model Validation

To validate the proposed model, we compared the results of the simulations given by the model to the experimental data. For this purpose, we have shown in Figure 9 the theoretical and experimental evolution of Re (Z) as a function of −Im (Z). A good fit is achieved between the experimental and the theoretical data calculated with a low error ($<10^{-3}$).

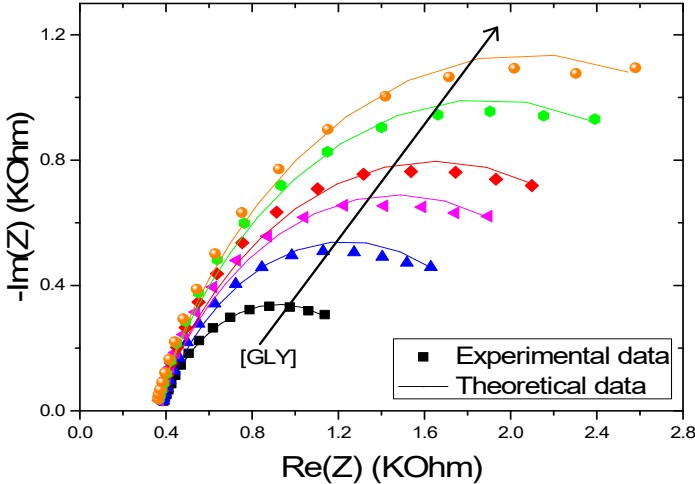

**Figure 9.** Experimental and theoretical electrochemical impedance spectroscopy (EIS) of CS-MIPs/ PPy/Au in 5 mM [Fe $(CN)_6]^{3-/4-}$ initial potential E = 0.2 V. f = 100 kHz–150 Hz. For different concentrations of GLY (0.31 pg/mL, 6.25 pg/mL, 125 pg/mL, 2.5 ng/mL, and 50 ng/mL).

### 3.3.4. Analysis of Theoretical Results

The various parameters determined by the model are gathered in Table 2. The thickness of the chitosan membrane is estimated at 88 μm. From the second incubation, a slight decrease in the latter was recorded (83 μm), which can be caused by the interpenetration of the chitosan in the polypyrrole surface. The resistance of the solution decreased from a value of 369.5 Ω to 348.5 Ω after the second incubation. This variation is related to the change in thickness. The value of the coefficient n predicted by the physical model is 0.71, indicating that the membrane of the MIPs is of porous morphology. This coefficient increased slightly (0.73) from the second measurement.

**Table 2.** Different parameters of CS-MIPs/PPy/Au estimated by the model for different concentrations of glyphosate (GLY).

| [GLY] | $l$ (μm) | $R_s$ (Ω) | $k°$ (μm/s) | $R_{ct}$ (Ω) | $Q.10^6$ ($S^n \, \Omega^{-1}$) | n | $e_1.10^4$ | $e_2.10^4$ |
|---|---|---|---|---|---|---|---|---|
| 0 | 88 | 369.5 | 55 | 1109.7 | 2.5 | 0.71 | 3.46 | 4.58 |
| 0.31 pg/mL | 88 | 369.5 | 48 | 1733.9 | 2.25 | 0.71 | 1.81 | 6.34 |
| 6.25 pg/mL | 83 | 348.5 | 39 | 2249.4 | 2.08 | 0.73 | 3.99 | 3.83 |
| 125 pg/mL | 83 | 348.5 | 32 | 2600.9 | 1.74 | 0.73 | 2.97 | 2.75 |
| 2.5 ng/mL | 83 | 348.5 | 27 | 3082.5 | 1.42 | 0.73 | 6.84 | 1.90 |
| 50 ng/mL | 83 | 348.5 | 24 | 3467.8 | 1.26 | 0.73 | 7.81 | 6.93 |

The simulated value of the standard rate of the electron transfer reaction k° on the characterized electrode before incubation in the GLY solution is equal to 55 μm/s. k° decreases with an increasing concentration of incubated GLY. This decrease is due to the occupation of the complementary cavities by GLY molecules, increasing the opposition to the transfer of electrons.

In Figure 10A, we have represented the evolution of the constant $k°$ as a function of the resistance $R_{ct}$ obtained from the model. $k°$ varies linearly as a function of $R_{ct}$ with a correlation coefficient $R^2 = 0.981$. The regression equation, thus, found is: $k° = -0.014\ R_{ct} + 70.72$.

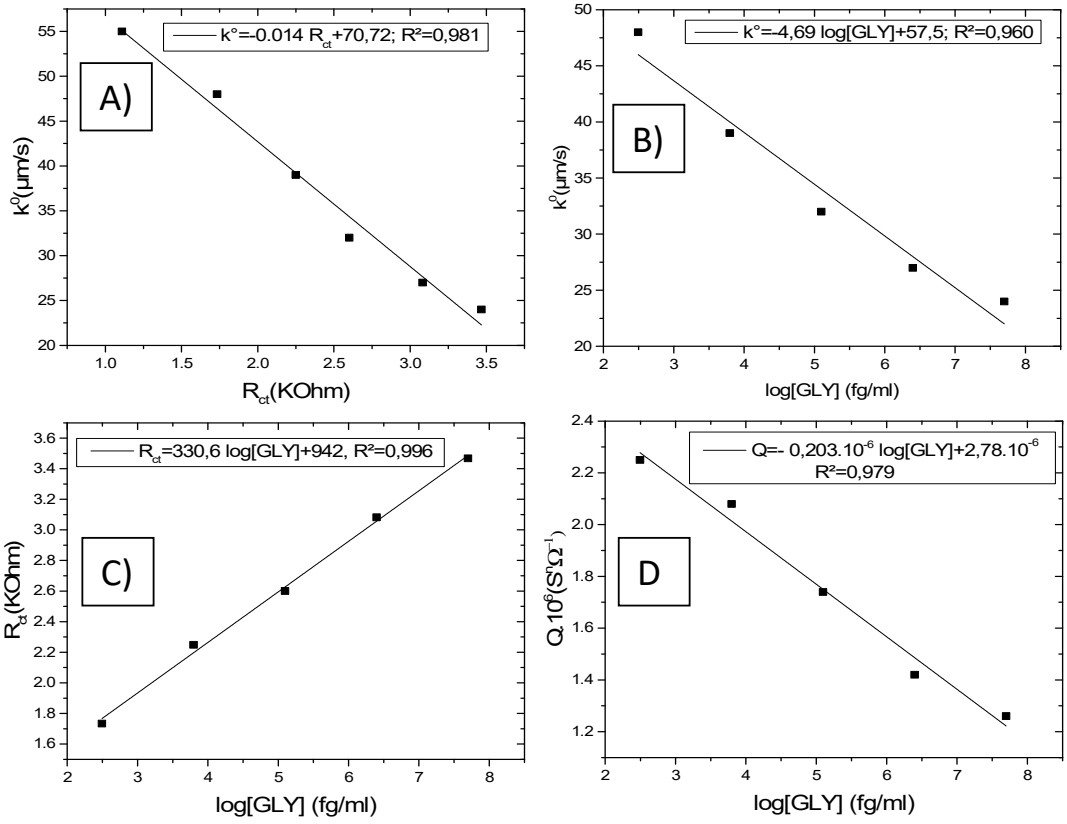

**Figure 10.** Calibration curves for CS-MIPs/PPy/Au electrode. (**A**) $k° = f(R_{ct})$ for different [GLY], (**B**) $k° = f(\log [GLY])$, (**C**) $R_{ct} = f(\log [GLY])$, (**D**) $Q = f(\log [GLY])$. ([GLY] = 0, 0.31 pg/mL, 6.25 pg/mL, 125 pg/mL, 2.5 ng/mL, and 50 ng/mL).

The variations of $k°$ (Figure 10B) and $R_{ct}$ (Figure 10C) as a function of the logarithm of the GLY concentration show excellent regression coefficient values. The linear regression equations are ($k° = -4.69 \log [GLY] + 57.5$; $R^2 = 0.960$) and ($R_{ct} = 330.6 \log [GLY] + 942$; $R^2 = 0.996$), respectively. This behavior is explained by the changes in the structure of the film by the occupation of the imprinted sites via the molecules of GLY, which makes it more resistive.

The values of the coefficient of CPE (Q) also decrease with the increase in the concentration of GLY, allowing the increase of the capacitive impedance that contributes to the change of the global impedance. This variation evolves according to a linear law ($Q = -0.203 \times 10^{-6} \log [GLY] + 2.78 \times 10^{-6}$; $R^2 = 0.979$) (Figure 10D). Q slightly varies with the GLY concentration, whereas $R_{ct}$ largely varies with the GLY concentration. It comes that the Faradic current is highly disturbed when GLY increases, which gives a high contribution to the global impedance whereas the capacitive current is slightly modified.

## 3.4. Model Exploitation for Optimization of the GLY Microsensor

### 3.4.1. Effect of Coefficient n

Using Equations (5) and (6), we can show the variation of the overall impedance for different values of the coefficient n (Figure 11). The variation of n from 0.6 to 1 shows an effect on the impedance spectrum, which increases proportionally. Over the same frequency range with increasing n, the impedance spectrum moves away from the *x*-axis in the high-frequency domain and approaches the

same axis in the low frequencies. Therefore, there is an optimal value of n that leads to a well-defined semicircle (0.8). Experimentally, n can be modified by varying the thickness of the MIPs' film by changing the number of cycles during the electrodeposition. It can be decreased by using a poro-gene solvent in the synthesis [30], which can increase the porosity and tortuosity on the film.

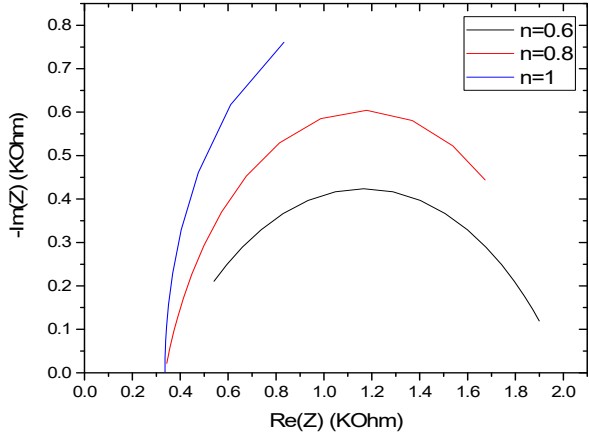

**Figure 11.** Effect of the coefficient n on the variation of the impedance response of CS-MIPs/PPy/Au for $[Fe(CN)_6]^{3-/4-}$ = 5 mM, f = 100 kHz–150 Hz. $k°$ = 0.005 cm/s, A = 0.0064 cm$^2$, Q = $10^{-6}$ s$^n\Omega^{-1}$, T = 298 K, $l$ = 80 μm.

### 3.4.2. Effect of Electron Transfer Rate Constant $k°$

The effects of varying the standard rate constant $k°$ on the microsensor response can be seen in Figure 12. $k°$ has no effect on the value of the impedance at the high frequency range. At the same time, the initial imaginary impedance increases with decreasing $k°$. Furthermore, the maxima of Re (Z) and −Im (Z) are inversely proportional to the increase in $k°$ and they shift toward the low frequencies. For obtaining a higher value of $k°$, the thickness of the film should be decreased. The concentration of the redox couple should be increased. The integration of conductive nanomaterials in the CS film could also increase the value of $k°$.

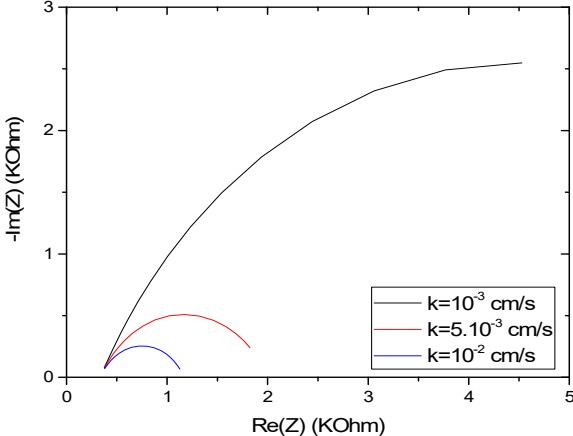

**Figure 12.** Effect of the rate constant $k°$ on the impedance response of CS-MIPs for $[Fe(CN)_6]^{3-/4-}$ = 5 mM, f = 100 kHz–150 Hz, n = 0.7, A = 0.0064 cm$^2$, Q = $10^{-6}$ s$^n\Omega^{-1}$, T = 298 K, $l$ = 80 μm.

### 3.4.3. Effect of the CPE Coefficient (Q)

In Figure 13, we have shown the impedance spectrum for different values of the CPE coefficient (Q). A well-defined semi-circle impedance spectrum is obtained for a given value Q, which is called optimal Q ($Q_{opt}$), $10^{-6}$ s$^n\Omega^{-1}$. For Q > $Q_{opt}$, −Im (Z) approaches zero, Re (Z) tends toward the value of

the resistance of the solution $R_s$. For $Q < Q_{opt}$, $-Im(Z)$ tends towards zero, and $Re(Z)$ tends to the value of the charge transfer resistance $R_{ct}$. Q can be varied by the variation of the concentration of the redox couple, the variation of the movement of the ions by agitation of the electrolyte, the variation of the temperature, the variation of the applied potential, or the changes in the morphology of the membrane [31–33].

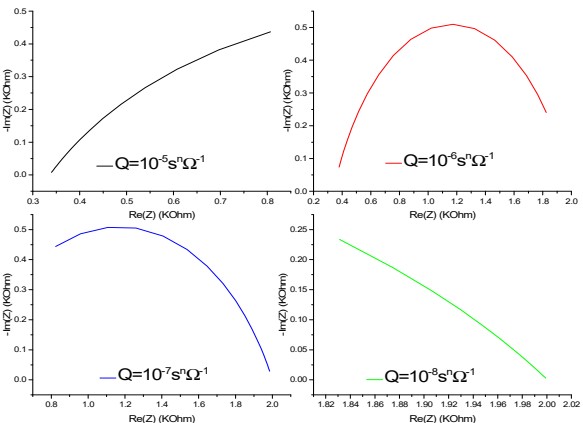

**Figure 13.** Effect of coefficient Q on the impedance response of CS-MIPs/PPy/Au for $[Fe(CN)_6]^{3-/4-}$ = 5 mM, f = 100 kHz–150 Hz, n = 0.7, A = 0.0064 cm$^2$, k° = 0.005 cm/s, T = 298 K, $l$ = 80 μm.

### 3.4.4. Effects of Membrane Thickness and Surface of CS-MIPs

From Equations (5) and (6), the thickness of the membrane ($l$) influences the resistance of the solution Rs only involved in the actual impedance equation. In Figure 14A, we have shown the variation of the impedance spectrum for different values of $l$. The maximum of $Re(Z)$ increases for greater thickness, which is explained by the change in the resistance of the solution. On the other hand, the maximum value of $-Im(Z)$ remains constant.

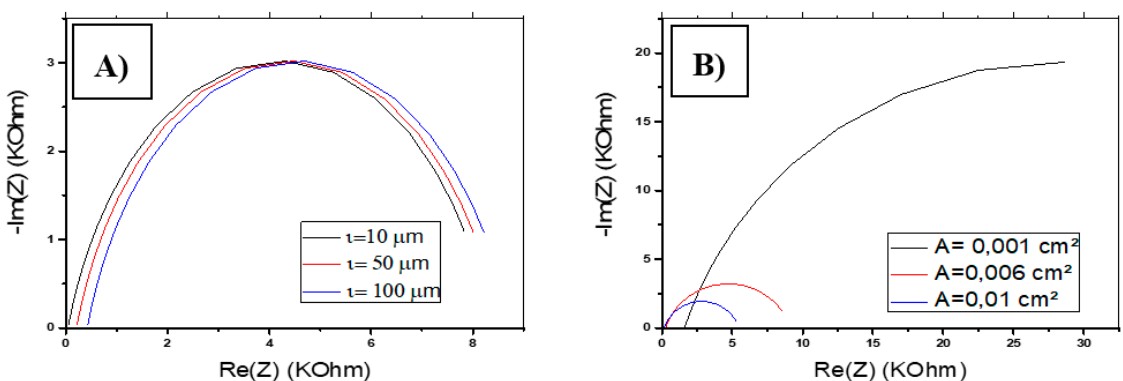

**Figure 14.** Variation of the impedance response of CS-MIPs/PPy/Au. (**A**) In terms of $l$ for $[Fe(CN)_6]^{3-/4-}$ = 5 mM, f = 100 kHz–150 Hz, n = 0.7, A = 0.0064 cm$^2$, k° = 0.005 cm/s, Q = $10^{-6}$ s$^n$Ω$^{-1}$,T = 298 K. (**B**) In terms of A for $[Fe(CN)_6]^{3-/4-}$ = 5 mM, f = 100 kHz–150 Hz, n = 0.7, A = 0.0064 cm$^2$, k° = 0.005 cm/s, Q = $10^{-6}$ s$^n$Ω$^{-1}$, T = 298 K, $l$ = 80 μm.

The variation of the membrane surface (A) strongly influences the overall impedance (Figure 14B). $Re(Z)$ and $-Im(Z)$ record larger values for a minimum area A. In addition, A affects the resistance of the solution, which tends toward 0 for a larger area A.

These variations caused by $l$ and A are the minimum that can be recorded on the overall impedance. In reality, $l$ and A can also influence the Faradic impedance by varying the rate of electron transfer expressed in units of distance per unit of time. They can also modify the capacitive impedance caused

by the change in membrane characteristics such as the coefficient n. The variation of $l$ and A modify the number of imprinted sites available in the film, which modifies the sensitivity of the sensor. In fact, larger A and $l$ are higher when the number of imprinted sites leads to a higher sensitivity of detection.

### 3.4.5. Effects of Temperature and Concentration of the Ferri/Ferrocyanide Solution

The higher temperatures of the ferri/ferrocyanide solution, used for the characterization of the electrode, cause an increase in the real and imaginary impedance, as shown in Figure 15A. The change in temperature shows an effect on the charge transfer resistance $R_{ct}$. However, it can also cause a change in the resistance of the solution $R_s$ by modifying its resistivity. According to this interpretation, it is recommended to work at a low temperature, which allows having a low charge transfer resistance.

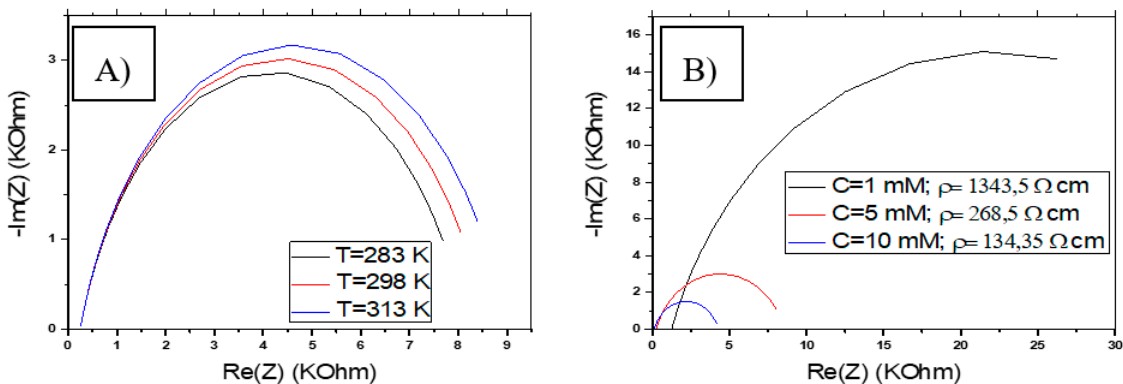

**Figure 15.** Variation of the impedance response of CS-MIPs/PPy/Au. (**A**) In terms of T for $[Fe(CN)_6]^{3-/4-} = 5$ mM, f = 100 kHz–150 Hz, n = 0.7, n = 0.7, k° = 0.005 cm/s, Q = $10^{-6}$ $s^n\Omega^{-1}$, A = 0.0064 cm², $l$ = 80 µm. (**B**) In terms of $[Fe(CN)_6]^{3-/4-}$ for f = 100 kHz–150 Hz, n = 0.7, k° = 0.005 cm/s, Q = $10^{-6}$ $s^n\Omega^{-1}$, A = 0.0064 cm², T = 298 K, $l$ = 80 µm.

The concentration of the ferri/ferrocyanide solution greatly influences the overall impedance Z. We have shown in Figure 15B, the variation of Z as a function of three concentrations of the redox couple (C = 1, 5, 10 mM). Increasing the concentration in the assay medium causes a decrease in Re (Z) and −Im (Z), including a decrease in $R_{ct}$. The concentration of the redox couple also influences the resistance of the solution by modifying its resistivity. A higher concentration leads to lower $R_s$. According to this interpretation, it is recommended to work at a high concentration ($\geq$5 mM), which means having a low charge transfer resistance.

The temperature and the concentration of the redox couple can also influence the rate of charge transfer characterized by its constant k°. These changes are due to the movement and density of electrons.

### 3.5. Analytical Performances of the CS-MIPs/PPy Functionalized Gold Electrode

The experimental conditions defined for the fabrication of the CS-MIPs/PPy functionalized gold electrode led to optimal values for n (0.7), Q ($<2.5 \times 10^{-6}$ $s^n\Omega^{-1}$), concentration of the redox (5 mM), measurements at room temperature, and too low of a value for k° ($<5.5 \times 10^{-3}$ cm/s), too low value of A (0.0064 cm²) for obtaining optimal impedance spectra. After the determination of the analytical performance, some ways for the improvement could be defined.

The relative variation of the charge transfer resistance of each electrode was then normalized using the following equation $|R_{ct}-R_{ct\ Blank}|/R_{ct\ Blank}$ ($\Delta R/R$). In Figure S2, the $\Delta R/R$ versus the logarithmic value of the GLY concentrations plot produced a linear relationship ranging from 0.31 pg/mL to 50 ng/mL with a correlation coefficient of 0.996. The limit of detection (LOD) of the considered sensor was estimated at 1 fg/mL. The analytical performance of the CS-MIPs/PPy/Au microsensor was compared to that of other MIP-based electrochemical sensors for the detection of GLY reported in the

literature (Table 3). To our knowledge, this microsensor had better performance compared to most of the previously reported sensors.

**Table 3.** A comparison of analytical parameters of the proposed CS-MIPs/PPy/Au microsensor with previously published sensors for glyphosate.

| Electrochemical Technique | Electrode | Linear Range | Limit of Detection | Reference |
|---|---|---|---|---|
| DPASV | MCA-MIPs-GNPs/PGE | 3.98–0.54 ng/mL | 0.35 ng/mL | [34] |
| SWV | PPy-MIPs/Au | 0.017 pg/mL–1.69 ng/mL | 0.17 pg/mL | [35] |
| DPV | PPy-MIPs-PB-HAuCl4/IOT | 400−1200 ng/mL | 92 ng/mL | [36] |
| EIS | CS-MIPs-PPy/Au | 0.31 pg/mL–50 ng/mL | 0.005 pg/mL | this work |

MCA-MIPs-GNPs/PGE: Molecularly Imprinted N-methacryloyl-L-cysteine- Gold nanoparticles/Pencil graphite electrode. PPy-MIPs/Au: Molecularly Imprinted Polypyrrole/Gold electrode. PPy-MIPs-PB-HAuCl4/IOT: Molecularly Imprinted Polypyrrole-Prussian blue-Urchin-like gold nanoparticles/Indium/tin oxide glass electrode. CS-MIPs-PPy/Au: Molecularly Imprinted chitosan-Polypyrrole/Gold micoelectrode. DPASV: Differential pulse anodic stripping voltammetry. SWV: Square wave voltammogram. DPV: Differential pulse voltammetry. EIS: Electrochemical Impedance Spectroscopy.

In order to assess the effectiveness of the imprinting, the detection of glyphosate is performed using CS-NIPs/Au. As reported in Figure 16, ΔR/R of the CS-MIPs/PPy/Au sensor is stronger than that of the CS-NIPs/PPy/Au sensor. The ratio of the sensitivities of MIP versus NIP is determined, leading to an imprinting factor of 11.5. This point indicates that the adsorption of GLY by the non-imprinted Chitosan is negligible and the effectiveness of the template imprinting is demonstrated.

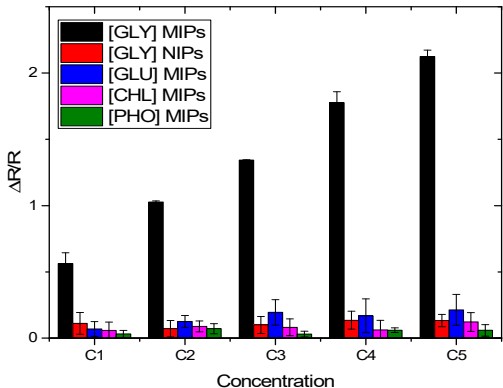

**Figure 16.** Selectivity of CS-MIPs/PPy/Au microsensor. (**black**) response of CS-MIPs/PPy/Au to GLY, (**red**) response of CS-NIPs/PPy/Au to GLY, (**blue**) response of CS-MIPs/PPy/Au to Gluphosinate-ammonium (GLU), (**pink**) response of CS-MIPs/PPy/Au to chlorpyrifos (CHL), (**green**) response of CS-MIPs/PPy/Au to phosmet (PHO).

The specificity of MIPs was tested with the detection of different pesticides that might be present in the same medium as GLY. Gluphosinate-ammonium (GLU), chlorpyrifos (CHL), and phosmet (PHO) were chosen to investigate the selectivity of this imprinted sensor (Figure 16). On the CS-MIPs/PPy/Au, ΔR/R of GLY was higher than that of the other three substances at the same concentration. The ratio of sensitivities were 32.6, 100, and 50 for GLU, CHLO, and PHO, respectively. Thus, we can conclude that the tested compounds will not interfere with the detection of GLY in the same concentration range, proving the selectivity of the MIP sensor.

To investigate the reproducibility of the CS-MIPs/PPy/Au microsensor, the experiment was performed using three individual electrodes, which were prepared in similar conditions. The results showed an acceptable reproducibility with a 1.29% relative standard deviation.

The CS-MIPs/PPy/Au microsensor was regenerated by incubating it in acetic acid/methanol solution (1:1, v/v) for 5 min to remove adsorbed GLY. The cycle was repeated five times (Figure S3). The second and the third cycles showed weak relative variation of the charge transfer resistance compared with the first cycle.

To evaluate the feasibility of the proposed sensor for its potential applications, the CS-MIPs/PPy/Au was used to determine GLY levels in a river water sample collected from the Rhone River in Lyon, France. EIS measurements were performed before and after incubation of the sensor in the Rhone river water for 30 min, and their respective Nyquist plots are almost superimposed, indicating that this sample does not contain GLY (Figure 17).

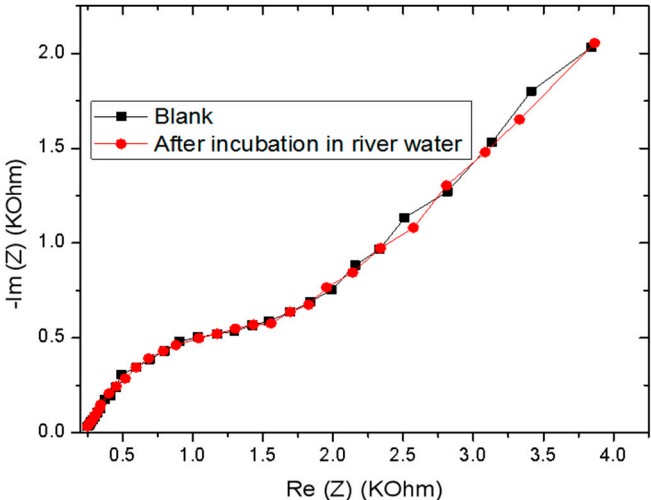

**Figure 17.** Electrochemical impedance spectroscopy (EIS) of CS-MIPs/PPy/Au before and after incubation in the Rhone river water.

To confirm the above observation, the standard addition method was used to detect GLY. This was done by gradually increasing GLY concentration in the sample. A remarkable variation of EIS was observed with the increase in GLY concentrations (Figure 18A). The variation of the charge transfer resistance is linearly proportional to the logarithmic value of the GLY concentrations in the range of 0.31 pg/mL to 50 ng/mL with $R^2$ equal to 0.986 (Figure 18B), indicating a good correlation between GLY concentration and the change of the impedance. The linear regression goes through zero ($\Delta R/R = 0.316 \log [GLY]$), which likely confirms the absence of GLY in the basic sample. Additionally, the sensitivity of this proposed sensor in the river water is the same as in the buffer, which revealed that the CS-MIPs/CMA/Au exhibited high recognition selectivity toward GLY in river water samples and almost without interference.

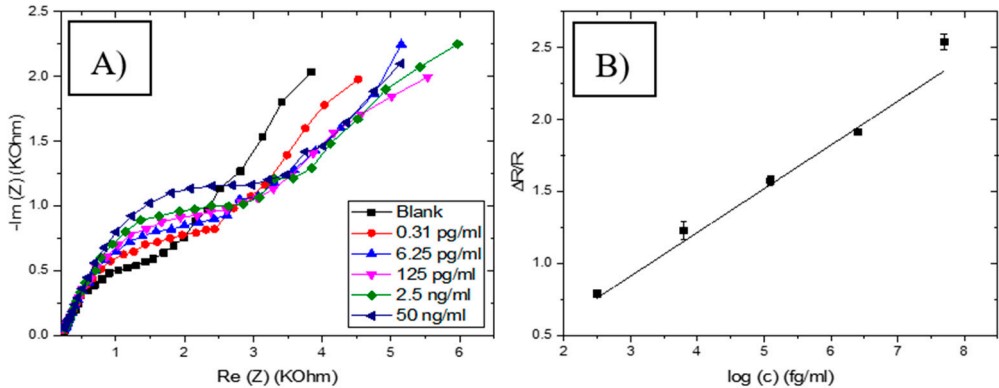

**Figure 18.** (**A**) Electrochemical Impedance Spectroscopy (EIS) of CS-MIPs/PPy/Au electrode in 5 mM $[Fe(CN)_6]^{3-/4-}$ and PBS before pre-concentration and after incubation in river water with different concentrations of GLY. (**B**) Detection of GLY in the Rhone river water using a standard addition method with the following concentrations (0.031 pg/mL, 6.25 pg/mL, 125 pg/mL, 2.5 ng/mL and 50 ng/mL) ($\Delta R/R = 0.316 \log [GLY]$, $R^2 = 0.986$).

## 4. Conclusions

In this work, a novel sensor has been constructed for the sensitive detection of glyphosate in water. It was based on electropolymerized polypyrrole films, doped with cobaltabis(dicarbollide) ions ($[3,3'\text{-Co}(1,2\text{-}C_2B_9H_{11})_2]$), as a solid contact layer between the gold microelectrode surface and the molecularly imprinted chitosan membrane for the sensitive detection of GLY (CS-MIPs/PPy/Au). Electrochemical Impedance Spectroscopy (EIS) was used for the selective detection of GLY between a wide range of concentration from 0.31 pg/mL to 50 ng/mL. EIS responses of the different micro-sensors were modeled by using mathematical modelization that described the phenomena at the electrode/electrolyte interface, while showing the effect of each parameter on the response signal, highlighting how GLY concentration and the experimental conditions can affect EIS parameters. The experimental conditions defined for the fabrication of the CS-MIPs/PPy functionalized gold electrode led to optimal values for n (0.7), Q ($<2.5 \times 10^{-6}$ $s^n\Omega^{-1}$), concentration of the redox (5 mM), measurements at room temperature. Some parameter values were found to be too low ($k° < 5.5 \times 10^{-3}$ cm/s, A = 0.0064 cm$^2$). After the determination of the analytical performance (LOD = 1 fg/mL), some ways for improvement could be defined in terms of sensitivity of detection.

**Supplementary Materials:** The following are available online at http://www.mdpi.com/2227-9040/8/4/104/s1. Figure S1: SEM images. (**A**) Bare gold, (**B**) PPy[3,3_-Co(1,2-C$_2$B$_9$H$_{11}$)$_2$]/Au, (**C**) CS-MIPs/PPy[3,3_-Co(1,2-C$_2$B$_9$H$_{11}$)$_2$]/Au, (**D**) CS-NIPs/PPy[3,3_-Co(1,2-C$_2$B$_9$H$_{11}$)$_2$]/Au. Figure S2: Sensitivity of CS-MIPs/PPy[3,3-Co(1,2-C$_2$B$_9$H$_{11}$)$_2$]/Au ($\Delta R/R = 0.30 \log [GLY] -0.15$; $R^2 = 0.996$). Figure S3: Regeneration of the CS-MIPs/PPy/Au microsensor.

**Author Contributions:** Methodology, A.A. and J.B. Software and validation, S.B.-B. Investigation, F.Z. Writing—original draft preparation, N.Z. Writing—review and editing, N.J.-R. Supervision, M.B. Funding acquisition, A.E. All authors have read and agreed to the published version of the manuscript.

**Funding:** The authors acknowledge the financial support of the EU H2020 research and innovation program entitled KardiaTool grant #768686 and from CAMPUS FRANCE program under grants PROFAS B+, PHC PROCOPE #40544QH, and PHC MAGHREB #39382RE.

**Conflicts of Interest:** The authors declare no conflict of interest.

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
