# Peer review of "Experimental Study and Mathematical Modeling of a Glyphosate Impedimetric Microsensor Based on Molecularly Imprinted Chitosan Film"

_chemosensors, doi:10.3390/chemosensors8040104_

Round 1

Reviewer 1 Report

Chemosensors-947259: Experimental study and mathematical modeling…

By F. Zouaqui, et al.

This manuscript describes another sensor targeted to glyphosate.  The herbicide has the been the focus of a number of sensing schemes due to its prevalence in weed control. The novelty of the reported device is cited by the authors as related to the use of a conductive polymer film beneath the actual capture/reactive imprinted chitosan film. The authors develop and report a mathematical model that appears to agree very closely with the experimental values.  The model is provided in great detail and the methodology of extracting parameters of importance to sensor functionality is well-documented. These author cited reasons for the publication of another sensor are all certainly valid and I recommend publication after several minor issues are addressed.

  • A quick Google and SciFinder search both turned up numerous other glyphosate sensors. A manuscript that reports a new device should both note the existence of these other devices and later compare the performance of the new sensor to those sensors that were already in the literature in order to justify any improvements that were observed.
  • It might be useful to the reader to expand the explanation of Figures 4. The discussion of the change in electron transfer ability is too brief and a somewhat unclear.

Author Response

The authors thank the reviewer for valuable comments that will improve the quality of the manuscript.

Reviewer #1 :

This manuscript describes another sensor targeted to glyphosate.  The herbicide has the been the focus of a number of sensing schemes due to its prevalence in weed control. The novelty of the reported device is cited by the authors as related to the use of a conductive polymer film beneath the actual capture/reactive imprinted chitosan film. The authors develop and report a mathematical model that appears to agree very closely with the experimental values.  The model is provided in great detail and the methodology of extracting parameters of importance to sensor functionality is well-documented. These author cited reasons for the publication of another sensor are all certainly valid and I recommend publication after several minor issues are addressed.

  • A quick Google and SciFinder search both turned up numerous other glyphosate sensors. A manuscript that reports a new device should both note the existence of these other devices and later compare the performance of the new sensor to those sensors that were already in the literature in order to justify any improvements that were observed.

A comparison of analytical parameters of the proposed CS-MIPs / PPy / Au microsensor with previously published sensors for glyphosate is presented in Table 3 and commented in lines 429-432.

  • It might be useful to the reader to expand the explanation of Figures 4. The discussion of the change in electron transfer ability is too brief and a somewhat unclear.

The comments about figure 4 were rewritten (lines 159-186), introducing quatitative data (RCT and Ia) for clarification. 

Reviewer 2 Report

The authors reported a glyphosate impedimetric microsensor based on molecularly imprinted chitosan film, and the mathematical modeling part was discussed. However, the investigation of the analytical performances was not sufficient.

Most importantly, the discussion about the selectivity of sensor was missing, which is an important part of the sensor fabrication. The authors should provide the performance of the sensor in the presence of the structure analog of the target glyphosate.

Secondly, the control experiment about the NIP electrode (prepared without the template) should be presented in detail. The NIP electrode should also be treated under the reported template removal condition (Line 135-137), and a compare of the NIP and MIP electrode in the electrochemical responses (Line 205) should be provided, in order to demonstrate the selectivity of the MIP sensor.

Moreover, the sensor should be applied in real samples, and the discussion about how the sensor avoids the non-specific adsorption should be provided.

Author Response

Reviewer #2 :

The authors reported a glyphosate impedimetric microsensor based on molecularly imprinted chitosan film, and the mathematical modeling part was discussed. However, the investigation of the analytical performances was not sufficient.

Most importantly, the discussion about the selectivity of sensor was missing, which is an important part of the sensor fabrication. The authors should provide the performance of the sensor in the presence of the structure analog of the target glyphosate.

 The discussion about selectivity of the sensor is presented in Figure 16 and the associated comments in lines 452-458.

Secondly, the control experiment about the NIP electrode (prepared without the template) should be presented in detail. The NIP electrode should also be treated under the reported template removal condition (Line 135-137), and a compare of the NIP and MIP electrode in the electrochemical responses (Line 205) should be provided, in order to demonstrate the selectivity of the MIP sensor.

  The discussion about the NIP electrode is presented in Figure 16 and the associated comments in lines 446-450.

Moreover, the sensor should be applied in real samples, and the discussion about how the sensor avoids the non-specific adsorption should be provided.

The application in the determination of glyphosate in real water samples is presented in lines 471-487 and Figures 17 and 18.

Round 2

Reviewer 2 Report

I have no other questions.